



**Hydrology and Earth System Sciences**

# Quantifying the impacts of land cover change on hydrological responses in the Mahanadi river basin in India

**Shaini Naha, Miguel A. Rico-Ramirez, and Rafael Rosolem**

Civil Engineering Department, University of Bristol, Bristol, BS8 1TR, UK

**Correspondence:** Shaini Naha (sn17546@bristol.ac.uk)

**Abstract.** The objective of this study is to assess the impacts of land cover change on the hydrological responses of the Mahanadi river basin, a large river basin in India. Commonly, such assessments are accomplished by using distributed hydrological models in conjunction with different land use scenarios. However, these models, through their complex interactions among the model parameters to generate hydrological processes, can introduce significant uncertainties to the hydrological projections. Therefore, we seek to further understand the uncertainties associated with model parameterization in those simulated hydrological responses due to different land cover scenarios. We performed a sensitivity-guided model calibration of a physically semi-distributed model, the Variable Infiltration Capacity (VIC) model, within a Monte Carlo framework to generate behavioural models that can yield equally good or acceptable model performances for subcatchments of the Mahanadi river basin. These behavioural models are then used in conjunction with historical and future land cover scenarios from the recently released Land-Use Harmonization version 2 (LUH2) dataset to generate hydrological predictions and related uncertainties from behavioural model parameterization. The LUH2 dataset indicates a noticeable increase in the cropland (23.3 % cover) at the expense of forest (22.65 % cover) by the end of year 2100 compared to the baseline year, 2005. As a response, simulation results indicate a median percent increase in the extreme flows (defined as the 95th percentile or higher river flow magnitude) and mean annual flows in the range of 1.8 % to 11.3 % across the subcatchments. The direct conversion of forested areas to agriculture (of the order of $30\,000\,\text{km}^2$) reduces the leaf area index, which subsequently reduces the evapotranspiration (ET) and increases surface runoff. Further, the range of behavioural hydrological predictions indicated variation in the magnitudes of extreme flows simulated for the different land cover scenarios; for instance, uncertainty in scenario labelled "Far Future" ranges from 17 to $210\,\text{m}^3\,\text{s}^{-1}$ across subcatchments. This study indicates that the recurrent flood events occurring in the Mahanadi river basin might be influenced by the changes in land use/land cover (LULC) at the catchment scale and suggests that model parameterization represents an uncertainty which should be accounted for in the land use change impact assessment.

## 1  Context and background

Land use/land cover (LULC) change induced by the rapid anthropogenic activities is one of the major causes of change in hydrological and watershed processes (Rogger et al., 2016). Alterations of existing land cover types and land management practices in a catchment can thereby significantly modify the rainfall path into runoff by changing the hydrological dynamics such as surface runoff, baseflow, evapotranspiration (ET), water holding capacity of the soil, interception and groundwater recharge, thus reflecting a change in the water demand (Berihun et al., 2019; Bosch and Hewlett, 1982; Costa et al., 2003; Foley et al., 2005; Garg et al., 2017; Hamman et al., 2018; Mao and Cherkauer, 2009; Rogger et al., 2016; Zhang et al., 2014). For instance, developing countries like India are facing rapid growth in population, which has prominent effects on LULC dynamics through deforestation, rapid urbanization and agricultural intensification, subsequently modifying the hydrological cycle in many river basins of India. A recent analysis on global land cover changes for the 2000–2017 period (Chen et al., 2019a; IPCC, 2019) revealed 86 % changes in land cover patterns in In-

dia with 82 % detected as croplands and the remaining 4 % as forests (Chen et al., 2019a; IPCC, 2019). Therefore, a comprehensive understanding and evaluation of land cover change impacts on hydrological processes are essential for decision makers to plan environmental policies which focus on water resource allocation, riparian ecosystem protection and river restoration (Chen et al., 2019b; Chu et al., 2013).

Many studies have attempted to evaluate the hydrological responses to different LULC patterns on specific geographic locations (Abe et al., 2018; Chu et al., 2013; Eum et al., 2016; Li et al., 2015; Ma et al., 2010; Rodriguez and Tomasella, 2016; Viola et al., 2014; Woldesenbet et al., 2017) including Indian river basins (Babar and Ramesh, 2015; Dadhwal et al., 2010; Das et al., 2018; Gebremicael et al., 2019; Wilk and Hughes, 2002). Most of these studies used physically distributed hydrological models (e.g., SWAT, VIC, MIKE-SHE) to simulate the complex hydrological processes and to examine the impact of LULC changes on those processes. Conventionally, this is done by calibrating and validating the hydrological model against the observed data and then setting up that single calibrated model for a baseline land cover scenario. The calibrated model is then run for different land use scenarios, and subsequently the differences in simulations are compared. However, it is widely recognized that hydrological predictions obtained from a single calibrated model can be biased; therefore, the measure of their reliability is always questionable (Beven and Binley, 1992; Huang and Liang, 2006). There may exist an "equally probable parameter set" that can yield equally good or acceptable model predictions (also known as behavioural models) which are identified due to the complex interactions among the model parameters to represent the complex hydrological processes. This is known as equifinality and is considered one of the main sources of uncertainty in hydrological modelling (Her et al., 2019). Recent climate change studies have acknowledged the uncertainties stemming from model parameters; therefore, they take into account these uncertainties while predicting the hydrological responses due to climate change (Chaney et al., 2015; Feng and Beighley, 2020; Her et al., 2019; Huang and Liang, 2006; Joseph et al., 2018; Mockler et al., 2016; Singh et al., 2014). However, little is known about the contributions of model parameter uncertainties to the land use change impacts; thus, very few studies exist (Breuer et al., 2006; Chen et al., 2019b) which reported that uncertainties associated with the model parameters could significantly influence land cover change impacts and hence should not be overlooked while modelling hydrologic responses to LULC change.

This paper specifically focusses on the Mahanadi river basin, an easterly flowing river basin in India. The eastern part of India is amongst the most rapidly changing landscape over the country; specifically, Mahanadi river basin has undergone drastic land cover changes in the last decades (Behera et al., 2018; Dadhwal et al., 2010). In this study, we address the following science questions:

1. What are the expected impacts of LULC changes on the water balance of the Mahanadi river basin?

2. How do these predicted impacts vary as a result of model parameter uncertainties?

The major objectives of this study are

1. to predict the changes in hydrological processes owing to historical and future changes in LULC and

2. to understand the contribution of uncertainty from hydrologic parameterization to the hydrologic projections due to LULC change.

To this end, a large-scale physically semi-distributed hydrological model, the Variable Infiltration Capacity (VIC) (Liang et al., 1994), and historical and future land cover scenarios from the Land-Use Harmonization 2 (LUH2) database (Hurtt et al., 2018) are used to simulate the discharge and other hydrological components at daily timescales in the Mahanadi river basin. The ability of VIC to simulate the impacts of LULC changes on hydrology are well documented in various research articles (Garg et al., 2017, 2019; Hurkmans et al., 2009; Mao and Cherkauer, 2009; Patidar and Behera, 2019; Zhang et al., 2014).

We first perform sensitivity analysis of the model parameters and calibrate the hydrological model using Monte Carlo simulations to identify behavioural model simulations that implicitly account for the uncertainties from model parameterization. Those behavioural models are then used to predict the hydrological impacts due to different LULC scenarios. The land cover scenarios used in this study are the most up-to-date scenarios, available from version 2 of the Land-Use Harmonization (LUH2) dataset, which represents future changes in the LULC based on Shared Socioeconomic Pathways (SSPs) and climate radiative forcing outcomes (Representative Concentration Pathways, RCPs) (Gidden et al., 2019). Previous studies (Breuer et al., 2006; Chen et al., 2019b) have focussed only on the historical land use scenarios to evaluate the hydrological impacts; however, and to our knowledge, this is the first study that uses applications of the VIC model in conjunction with future land cover datasets produced under combined SSP and RCP scenarios. While most past studies in other catchments used aggregated (monthly) time steps to model the change, we use daily time steps to capture the dynamics of daily flow variability. Moreover, analysis carried out in most land use impact studies are typically limited to the streamflow, missing an overall picture of the hydrological processes.

Hydrol. Earth Syst. Sci., 25, 1–19, 2021                                    https://doi.org/10.5194/hess-25-1-2021

## 2 Research area

### 2.1 Geographical overview

The Mahanadi river basin is located in the eastern part of India (Fig. 1) and drains an area of 141 589 km², which nearly accounts for 4.3 % of the total geographical area of India. The basin has a varying topography with its lowest elevated area (−17 m) lying in the coastal reaches and the highest elevated area (1323 m) in the northern hills. The basin is characterized by tropical climate zone and receives rainfall from southwest monsoons which commence in June and last till October. The average annual rainfall is 1200 mm, with 90 % of the total annual rainfall occurring during the monsoon months (Jin et al., 2018). The mean annual discharge is 1895 m³ s⁻¹. The basin is also subjected to spatial variability in terms of receiving rainfall which has resulted in floods in some parts of the basin and drought in others. Notice that about 65 % of the basin is placed upstream of the Hirakud dam. The Hirakud dam with a gross storage capacity of 8.136 km³ is the major hydro-project in the river basin constructed in the year 1957 to alleviate the flood problems and to serve multiple other purposes such as irrigation, hydropower generation and supplying drinking water. Despite its significant storage capacity, the large flows from upstream of the dam and middle reaches of the catchment cause devastating floods during the monsoon in the deltaic region of the basin.

About 48 % of the total area is under agriculture, out of which 30 % is cropped during the kharif season or monsoon (June–October), and 15 % is under double or triple irrigation. The remaining 3 % of the area is cropped during rabi and zaid seasons (winter and summer, respectively). Deciduous Broadleaf Forest (DBF) being dominant among other forest types, covers 25 % of the basin area (Fig. 2a). Built up, plantation, grassland, shrubland, water bodies and other forest types constitute the rest (22 %) of the basin area. Comparison of the local historical LULC maps of 2005 and 2014, obtained from the National Remote Sensing Centre (NRSC), shows an increase in the agricultural land from about 43 % to 48 % at the expense of fallow land, built up areas and water bodies, while changes in forest covers were insignificant. In addition, loamy and clayey are the major soil types covering roughly 53 % and 42 %, respectively, of the total basin area (NBBSS-LUP, India). Approximately 90 % of the basin has moderately shallow to deep soil, having depths greater than 50 cm.

## 3 Materials and methods

### 3.1 Variable Infiltration Capacity (VIC) model

The VIC model is a semi-distributed, land surface hydrologic model which solves both water and energy balances within the grid cells (Cherkauer and Lettenmaier, 1999). VIC maintains sub-grid heterogeneity in land cover classes, i.e., divides each grid into tiles based on the number of land cover classes, and also considers sub-grid variability in the soil moisture storage capacity (Liang et al., 1994). Surface runoff in VIC is generated through an infiltration excess by using the Xiangjiang formulation (Zhao et al., 1980) in the upper two soil layers. Baseflow is generated from the third soil layer by applying the Arno formulation (Franchini and Pacciani, 1991). Actual evapotranspiration of each grid cell in VIC is obtained by summing up three types of evaporation: evaporation from bare soil, evaporation from canopy layer for each vegetation type and transpiration from different vegetation types; it is then weighted by the fractional area of each vegetation class. VIC computes potential evapotranspiration using the Penman–Monteith equation. The amount of rainfall intercepted by the canopy is calculated as a function of leaf area index (LAI).

To obtain the discharge at the basin outlet, the VIC model is coupled to a stand-alone routing model (Lohmann et al., 1996). This routing model follows a simple river routing scheme where runoff and baseflow are first routed to the edge of the grid cells using an instantaneous unit hydrograph and finally transported to the river/channel network using a linearized St. Venant equation. More details about the structure and formulations of the model can be found in the literature (Gao et al., 2010; Liang et al., 1994).

In this study, we implement the VIC model with three soil layers (known as VIC-3L), version 4.2.d in the water balance mode at a daily time step and at a grid resolution of 0.05° over the five subcatchments of the Mahanadi river basin. Note that the VIC model is commonly employed at daily scales especially when running with the water balance mode only (Gou et al., 2020; Hengade et al., 2018; Hurkmans et al., 2009). Flows are routed to the subcatchments of Basantpur (Ba), Kantamal (Ka), Kesinga (Ke), Sundergarh (Su) and Salebhata (Sa) (Fig. 1). We abstained from routing the flow for the entire Mahanadi river basin due to the presence of a major water management structure, Hirakud dam, at the middle reach of the basin.

### 3.2 Datasets

The key input data required by the VIC model are meteorological forcings (precipitation, maximum temperature, minimum temperature and wind speed), soil type, land cover information and topographic features. Topographical features are determined using the 30 m CARTO-DEM (Cartosat-1 digital elevation model), a national DEM developed by ISRO (Indian Space Research Organization) (Sivasena Reddy and Janga Reddy, 2015). The Mahanadi river basin is delineated and is converted into grid format of resolution 0.05° constituting 4807 grids within the basin area. Daily gridded precipitation (resolution 0.25°) and maximum and minimum temperature (resolution 1°) for the time period 1988–2010 are obtained from India Meteorological Department (IMD) (Pai et al., 2014). Soil textures are derived from the digitized

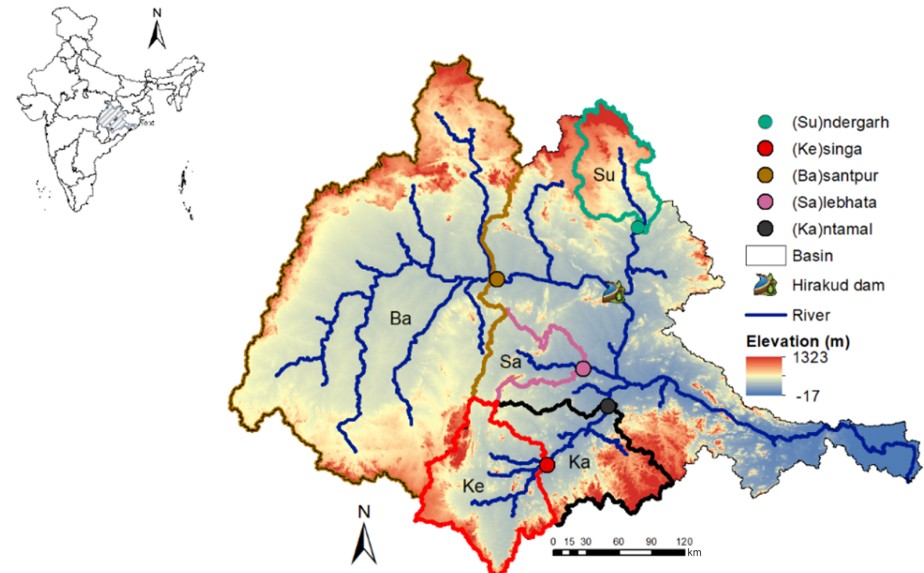

**Figure 1.** The Mahanadi river basin boundary and the analysed flow gauges and their catchments. Abbreviations for catchment names are Ba – Basantpur, Ka – Kantamal, Ke – Kesinga, Su – Sundergarh and Sa – Salebhata.

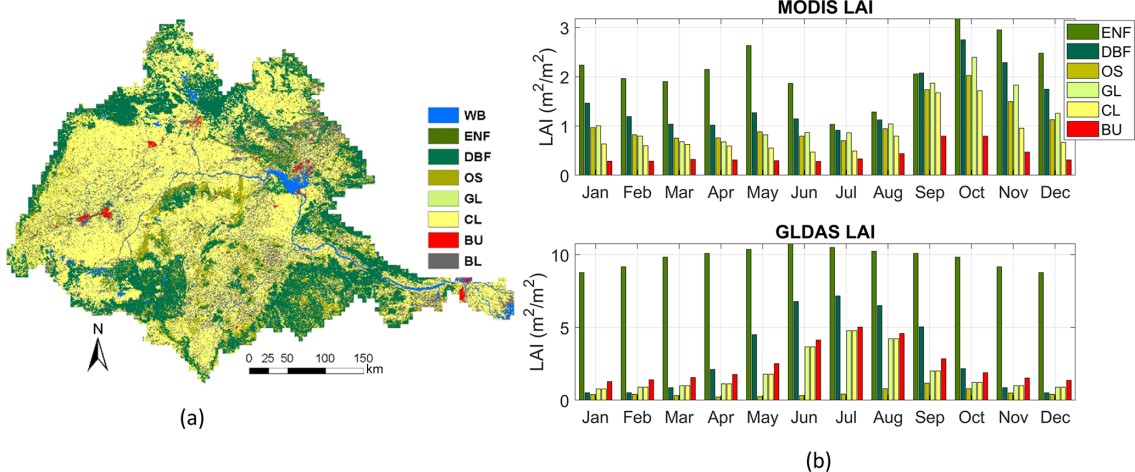

**Figure 2. (a)** LULC map of Mahanadi river basin from NRSC of year 2005. **(b)** Comparison of LAI values from MODIS, averaged over the time period 2000–2015, and GLDAS.

soil map as provided by the National Bureau of Soil Survey and Land Use Planning (NBSSLUP) (scale 1 : 250 000). Land cover maps from two different sources, i.e., local and global, are used in this study. The local LULC map is derived from National Remote Sensing Centre (NRSC), India, of year 2005 (scale 1 : 250 000; resolution 56 m) and is used in the model runs while performing sensitivity analysis, model calibration and validation. Global land cover scenarios are obtained from LUH2, which are used in model simulations for predicting impacts of land cover changes on hydrological components. All LULC maps used in this study are reformatted and reclassified into United States Geological Survey (USGS) LULC types as required by the VIC model

(Fig. 2a). The observed discharge at daily scales at multiple gauges (Fig. 1) for the simulated time (1988–2010) are obtained from the Central Water Commission (CWC), India, for validating the simulated discharge.

### 3.3 Model parameters

We have selected 16 VIC model parameters (Table 1) for the sensitivity analysis (SA). The choice of parameters was based on our preliminary experiments and expected sensitive properties from previous studies (see description below Table 1). Typical calibration in VIC involves only streamflow-related parameters as also recommended by VIC model de-

velopers (Gao et al., 2010; Gou et al., 2020; Xie et al., 2007). However, a few studies have reported that some vegetation parameters are sensitive to the runoff in the VIC model (Demaria et al., 2007; Joseph et al., 2018). Parameters subjected to SA in this study include, among others, rarely implemented soil properties, such as bulk density (BD) and fractional water content at wilting point ($wp_f$) and at critical point ($wcr_f$); vegetation properties, such as architectural resistance ($r_{arc}$) and stomatal resistance ($r_{min}$); and routing parameters, such as velocity ($v$) and diffusion (diff). A multiplier of $wcr_f$ is used to compute $wp_f$ to meet the criteria that soil moisture at wilting point should always be less than soil moisture at critical point, and the multiplier is tested for sensitivity rather than the actual parameter. A similar approach is followed by Rosolem et al. (2012) while testing sensitivity of parameters in a land surface model. Feasible ranges (minimum and maximum values) of soil parameters (BD, $wcr_f$, ksat, Exp) are obtained based on average hydraulic properties of USDA soil textural classes (Cosby et al., 1984; Rawls et al., 1998; Reynolds et al., 2000) considering only the dominant soil textures within the basin. Ranges for the rest of the soil parameters are based on suggestions from the VIC model developers and published studies. Feasible ranges of the vegetation parameters are obtained based on the recommended ranges provided in the Land Data Assimilation System (LDAS) values for the dominant vegetation types in the basin. Our preliminary experiments suggest canopy height is not sensitive; hence, roughness length (RL) and displacement height (Disp), which are computed from canopy height, are not accounted for in SA.

In addition, the LAI is an important vegetation factor, having substantial control over the water balance by directly influencing the ET rates (Gao et al., 2010; Matheussen et al., 2002). LAI is specified at a mean monthly basis in VIC. We compared the monthly-mean LAI averaged over the time period 2000–2015 from MODIS (Moderate Resolution Imaging Spectroradiometer) Aqua/Terra with the LAI values from the GLDAS (Global Land Data Assimilation System) database for the river basin. We observed that the monthly-mean LAI of all the LULC types from MODIS captures the phenological characteristics more realistically than the GLDAS LAI (Fig. 2b), which shall have further implications on water balance. We find that the range of MODIS LAI obtained for each LULC type are well in agreement with the LAI values obtained in the nearby Ganga river basin in India (Patidar and Behera, 2019).

Another important factor linking vegetation characteristics to hydrological processes in VIC is the root zone distribution. Typically, root zone allocation in VIC requires user-defined root zone depths and fractions for each land cover type that are kept fixed during the calibration process. We derived root zone depths and estimated the fractions of roots in each zone following Zeng (2002) for each vegetation type, and we used a simplified approach to vary the root zone distributions with respect to the soil depths during calibration. This ensures root

zone properties vary for different model calibration with a reduced number of parameters, hence providing a more manageable calibration strategy. For details on our root allocation approach, please refer to the Supplement (Sect. S1).

### 3.4 Experimental design

#### 3.4.1 Morris method for sensitivity analysis

SA of the chosen 16 VIC-3L parameters (Table 1) is conducted using the Morris (1991) method. This method requires Monte Carlo simulations where the model is run with a specified number of samples and measures the change in the model output by varying one parameter at a time. We used the one-at-a-time Latin hypercube sampling (LHS-OAT) strategy to form a total number of 1200 model parameter sets. This method proposed two sensitivity measures: (1) the mean ($\mu$) of the elementary effects, which estimates the direct effect of the input parameter on model output, and (2) the standard deviation ($\sigma$) of the elementary effects, which estimates the interaction between the input parameters on the model output. We tested the sensitivity of model parameters on the Kling–Gupta efficiency (KGE) metric (Eqs. 1–3) (Gupta et al., 2009), computed using observed daily streamflow values over 20 years (1990–2010) of simulation period.

$$KGE = 1 - \sqrt{(r-1)^2 + (\alpha-1)^2 + (\beta-1)^2};\qquad(1)$$

$$\alpha = \frac{\sigma_{sim}}{\sigma_{obs}},\qquad(2)$$

$$\beta = \frac{\mu_{sim}}{\mu_{obs}},\qquad(3)$$

where $r$ is the linear correlation between observed and simulated discharge, $\alpha$ is an estimate of flow variability error and $\beta$ is a bias term. $\sigma_{sim}$ and $\sigma_{obs}$ are standard deviations in simulated and observed discharge, respectively. $\mu_{sim}$ and $\mu_{obs}$ are mean of simulated and observed discharge, respectively.

We first visually inspect SA results and assume a screening threshold value for the sensitivity index, below which the parameters can be regarded as either completely insensitive or less influential. This is a common practice followed in previous SA studies (Gou et al., 2020; Sarrazin et al., 2016; Tang et al., 2007; Vanrolleghem et al., 2015). Next, to achieve a more objective screening convergence result, we compute the width of the 95 % confidence interval of the sensitivity indices (Herman et al., 2013; Wang and Solomatine, 2019) and then use maximum width of the 95 % confidence interval, as a statistic (Sarrazin et al., 2016), across the lower influential input to verify if the screening convergence has been reached. For a detailed explanation about the steps we took for the SA experiments, please refer to the Supplement (Sect. S2).

#### 3.4.2 Model calibration and validation

Next, we calibrate sensitive parameters separately on a sub-basin level for the time 1990–2000 with a 2-year warm-

**Table 1.** VIC and routing model parameters tested for sensitivity analysis and feasible ranges.

| Parameters | Description | Units | Minimum | Maximum |
|---|---|---|---|---|
| Soil parameters | | | | |
| $wcr_f$ | Fraction of water content at critical point[b] | – | 0.40 | 0.60 |
| $wp_f^c$($wp_f = M \cdot wcr_f$) | Fraction of water content at wilting point[b] | – | 0.50 | 0.99 |
| BD | Bulk density of soil (used in VIC estimation of porosity)[b] | $kg\,m^{-3}$ | 1350 | 1550 |
| ksat | Saturated hydraulic conductivity[b] | $mm\,d^{-1}$ | 240 | 840 |
| Exp | Parameter characterizing the variation of saturated hydraulic conductivity with soil moisture[b] | – | 10 | 30 |
| $d1$ | Thickness of first soil layer[a] | m | 0.01 | 0.3 |
| $d2$ | Thickness of second soil layer[a] | m | 0.31 | 3.5 |
| $d3$ | Thickness of third soil layer[a] | m | 0.31 | 3.5 |
| **dsmax** | **Max velocity of baseflow** [a] | **$mm\,d^{-1}$** | **$10^{-4}$** | **$10^{1.48}$** |
| **ds** | **Fraction of max velocity of baseflow** [a] | **–** | **$10^{-4}$** | **$10^{0}$** |
| **binf** | **Parameter to describe the Variable Infiltration Curve** [a] | **–** | **$10^{-4}$** | **$10^{0.6}$** |
| **ws** | **Fraction of maximum soil moisture of the third layer** [a] | **–** | **$10^{-4}$** | **$10^{0}$** |
| Vegetation parameters | | | | |
| $r_{arc}$ | Architectural resistance[b] | $s\,m^{-1}$ | 20 | 70 |
| $r_{min}$ | Minimum stomatal resistance[b] | $s\,m^{-1}$ | 100 | 170 |
| routing | | | | |
| $v$ | Flow velocity[b] | $m\,s^{-1}$ | 0.1 | 3 |
| diff | Flow diffusivity[b] | $m^2\,s^{-1}$ | 500 | 5000 |

Parameter names in bold are sampled on log domain. [a] indicates parameters that are suggested by VIC model developers as the most sensitive parameters (Gao et al., 2010). [b] indicates parameters suggested in the literatures to be tested for sensitivity (Demaria et al., 2007; Gou et al., 2020; Joseph et al., 2018; Yanto et al., 2017). [c] $wp_f$ is analysed based on its multiplier (i.e., the $M$ term in $wp_f$ parameter's equation). Although description and units refer to actual parameter in VIC, parameter range represents the multiplier values (instead of actual parameter).

up period (1988–1999), using a sequence of Monte Carlo simulation, by generating 5000 near-random parameter sets from within the specified range using the Latin hypercube sampling method (LHSM) with uniform distribution. We use KGE (Eq. 1) as the objective function to assess the model performance in the calibration period. The KGE metric balances the contribution to the error coming from all three main components, namely correlation (e.g., timing/dynamics), variability (e.g., seasonality), and systematic bias, and it is now a widely used metric in hydrometeorological studies (Gupta et al., 2009; Knoben et al., 2019). KGE ranges in $[-\infty, 1]$ with larger values indicating better performance. Additionally, we use the percent bias (PBIAS) to evaluate our model performance, especially to account for the high flow conditions. We adopt a common practice of se-

lecting the best model simulations by using a top certain percentage of the total simulations (Chaney et al., 2015; Mockler et al., 2016). This is relevant in our study as choosing model simulations based on a particular KGE score is subjective given that the behavioural performance, as well as the behavioural parameters, vary across the subcatchments. Therefore, we first assess the performance of top 10 %, 5 % and 2 % of model simulations at every subbasin and choose the top 2 % based on overall model performance across the subcatchments, hence not compromising the performance quality and also accounting for equifinality. These behavioural models are further used to simulate streamflow in the validation period (2001–2010) for all the subcatchments.

### 3.4.3 LULC scenarios

All the simulations in the calibration and validation period are performed using a static local LULC map of year 2005 derived from NRSC. Simulations using this land use map shall be termed NRSC2005 henceforth. Next, we used a set of land use scenarios based on Shared Socioeconomic Pathways (SSPs) and Representative Concentration Pathways (RCPs) from the recently released Land-Use Harmonization project (LUH2) dataset (releases LUH2v2h and LUH2v2f) for the time periods of 850–2005 and 2015–2100, respectively (Hurtt et al., 2018) (see Table S2, Supplement). The LUH2 approach estimates the gridded land use fractions, annually at a resolution of 0.25°. The land use fraction maps are available for each land use type at a resolution of 0.25°. So, we have first obtained LUH2 fraction maps of different LULC types for the Mahanadi basin extent at a resolution of 0.25° and further regridded to VIC grid size of 0.05°. Next, to run the VIC model, we have prepared a vegetation parameter file where we included the fractional coverage of all LULC types for each grid cell ensuring that each grid will contain more than one vegetation type. The land use classes are reduced to simplify our model application and consequently remapped to the VIC land use classes by assuming all primary (forested or non-forested) and secondary (forested and non-forested) land to Deciduous Broadleaf Forest (DBF); managed pasture and rangeland are considered grassland, and all crops are merged into a single class labelled "Cropland". Urban land and water bodies are retained (see Table S3, Supplement). It is worth mentioning that the "potentially non-forested secondary land" class in the LUH2 datasets matched to the forested areas in NRSC2005 and hence both mapped into DBF, which is the dominant forest type in the basin (Fig. S5 in the Supplement).

We used the behavioural models to simulate discharge for the baseline scenario using land cover map from LUH2 of year 2005 so as to attain more confidence in the future scenarios. We compare LULC maps, NRSC2005 and LUH2005 (Fig. 3) and observe spatial patterns of the most dominant land use classes; classes Cropland (CL) and Forest (F) show a similar spatial distribution and have comparable aerial coverage. The only notable difference in both maps is that the Barren Ground (BG) class is missing in LUH2005. Table 2 shows the percentage of area covered by each land use class in the basin. Note that we will refer to DBF as Forest (F) henceforth.

Among the future scenarios, owing to the large computational demand of our simulations, we only considered the worst case scenario, RCP3.4 SSP4, which resulted in maximum change in the land cover fractional area (Fig. 4). For our study, we have not taken into account the actual uncertainty due to the land cover scenarios. However, the percentage of land cover change relative to the baseline from other LUH2 scenarios is either negligible or comparable to our chosen scenario. Therefore, our chosen scenario which

**Table 2.** Percent of each land use type in NRSC2005 and LUH2005 in the entire Mahanadi river basin (WB – Water Body; ENF – Evergreen Needleleaf Forest; DBF – Deciduous Broadleaf Forest; GL – Grassland; CL – Cropland; U – Urban; BG – Barren ground).

| LULC classes | NRSC2005 (%) | LUH2005 (%) |
|---|---|---|
| WB | 2.60 | 0.76 |
| ENF | 0.08 | 0.00 |
| DBF | 35.98 | 41.00 |
| GL | 0.13 | 4.70 |
| CL | 49.00 | 53.00 |
| U | 0.52 | 0.40 |
| BG | 12.30 | 0.00 |

shows the maximum changes in land cover will likely produce the largest impact.

Land cover changes and fractional area covered in other future scenarios are shown in Fig. S6 in the Supplement. Four distinct years (i.e., four distinct land cover maps) have been chosen for this study: 2005 (Baseline), 2015 (Present), 2050 (Near Future) and 2100 (Far Future) to study the impacts of LULC change in the Mahanadi river basin. A sharp decrease in the forest cover is observed at the expense of agriculture in the years 2050 and 2100 (Fig. 4). We run the behavioural models three times using the individual LUH2 datasets: (1) with land use map "LUH2015", termed as the "Present" (P) scenario; (2) with land use map "LUH2050", termed as the "Near Future" (NF) scenario; and (3) with land use map "LUH2100", which is termed as the "Far Future" (FF) scenario. To account for the extreme hydrological effects that these changes could cause, two hypothetical scenarios are framed: (1) the "All Cropland" (CL) scenario where all the grassland and forest areas are transformed into cropland and (2) the "All Forest" (F) scenario where all the cropland and grassland areas are transformed into forest. The urban and water bodies in these hypothetical scenarios are retained as per the baseline scenario. Notice that the daily meteorological forcing used in all the model simulations is the same and obtained from the current climatology (i.e., 1990–2010). Here, we focus on identifying the impacts on hydrological responses mainly by applying individual land cover scenarios. Therefore, any changes observed in the predicted hydrological components will be only attributed to changes in LULC. It is also worth mentioning that running model simulations with different land cover scenarios would not directly impact the soil parameters identified in our chosen behavioural models. That is because all soil-related parameter values in VIC are assigned solely based on soil textures. The percent areas covered by each land use class at all subcatchments across the scenarios are shown in Table 3.

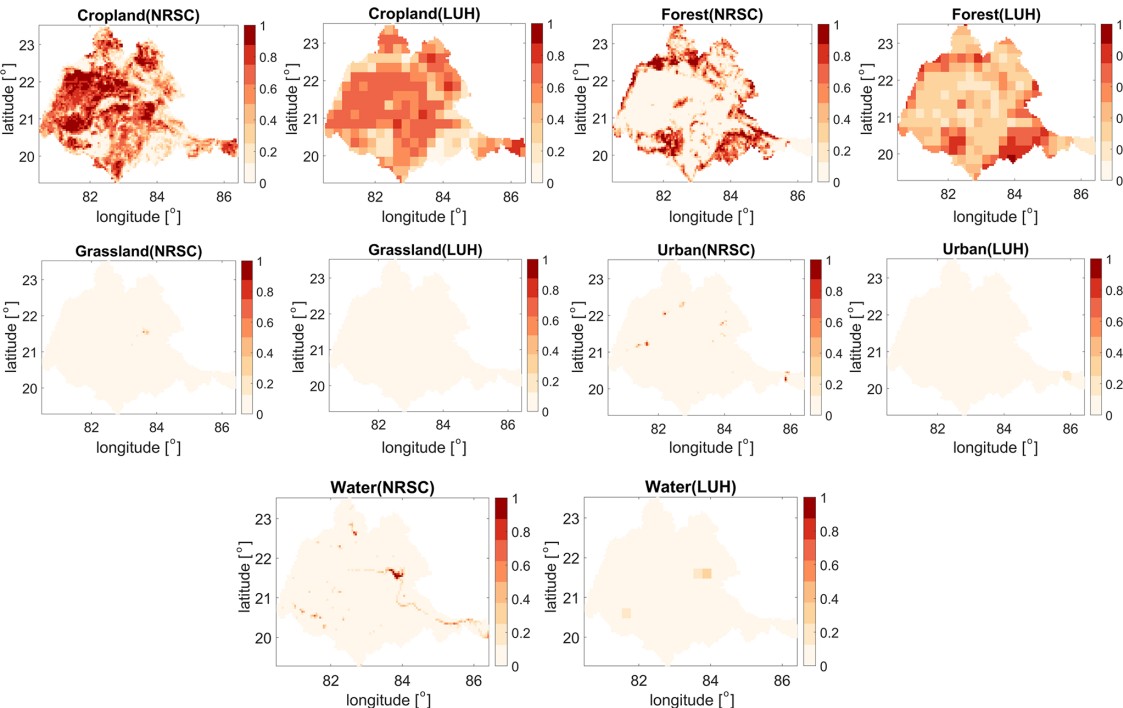

**Figure 3.** Comparison of spatial patterns of land cover types from NRSC and LUH2 for the baseline year, 2005. All land cover classes shown here are resampled to the model grid resolution of 5 km. The colour bar represents the fraction of area covered by each land cover type.

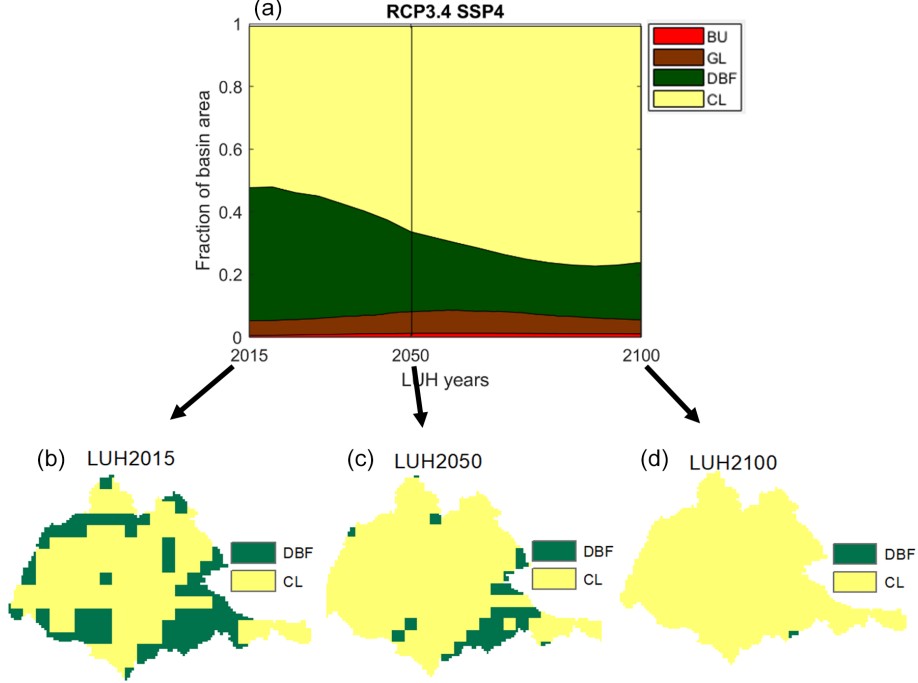

**Figure 4. (a)** Fraction of catchment area occupied by land use classes for scenario RCP3.4 SSP4. **(b–d)** Land cover scenarios from LUH2 (resolution – 25 km) for years 2015, 2050 and 2100 used in this study. LUH2 land cover classes shown here are resampled to the model grid resolution, and only the predominant class is shown here for clarity. For actual model simulations, VIC accounts for the individual proportion for each land cover type at each grid point.

**Table 3.** Land cover area change across all subcatchments of the Mahanadi river basin.

| LULC classes (%) | Baseline 2005 | Present 2015 | Near Future 2050 | Far Future 2100 | All Cropland | All Forest |
|---|---|---|---|---|---|---|
| | | | Basantpur | | | |
| CL | 40 | 54 | 69 | 78 | 94 | 0 |
| F | 54 | 41 | 23 | 16 | 0 | 94 |
| GL | 1 | 4 | 6 | 4 | 0 | 0 |
| WB | 5 | 1 | 1 | 1 | 5 | 5 |
| U | 1 | 1 | 1 | 1 | 1 | 1 |
| | | | Kantamal | | | |
| CL | 51 | 44 | 58 | 70 | 95 | 0 |
| F | 44 | 51 | 33 | 25 | 0 | 95 |
| GL | 0 | 5 | 8 | 5 | 0 | 0 |
| WB | 5 | 0 | 0 | 0 | 5 | 5 |
| U | 0 | 0 | 1 | 1 | 0 | 0 |
| | | | Kesinga | | | |
| CL | 44 | 50 | 62 | 73 | 95 | 0 |
| F | 51 | 45 | 30 | 22 | 0 | 95 |
| GL | 0 | 5 | 7 | 5 | 0 | 0 |
| WB | 5 | 0 | 0 | 0 | 5 | 5 |
| U | 0 | 0 | 1 | 1 | 0 | 0 |
| | | | Sundergarh | | | |
| CL | 29 | 67 | 77 | 83 | 96 | 0 |
| F | 67 | 29 | 17 | 15 | 0 | 96 |
| GL | 0 | 3 | 4 | 2 | 0 | 0 |
| WB | 4 | 0 | 0 | 0 | 4 | 4 |
| U | 0 | 0 | 1 | 1 | 0 | 0 |
| | | | Salebhata | | | |
| CL | 34 | 61 | 73 | 83 | 95 | 0 |
| F | 61 | 34 | 19 | 11 | 0 | 95 |
| GL | 0 | 0 | 7 | 6 | 0 | 0 |
| WB | 5 | 0 | 0 | 0 | 5 | 5 |
| U | 0 | 0 | 1 | 0 | 0 | 0 |

## 4 Results

### 4.1 Sensitivity analysis, model calibration and validation

It is to be noted that SA is conducted for all subbasins individually; hence, the Morris screening results obtained for each subbasin are independent of each other. However, we observe that the non-influential parameters match closely with each other across subbasins (Fig. S2). Based on the Morris sensitivity measures, there are six sensitive (or influential) parameters, namely dsmax, $d2$, binf, $v$, ws and ds. The rest of the parameters ($r_{min}$, $d3$, wcr$_f$, wp$_f$, $r_{arc}$, Exp, BD, diff, $d1$, ksat) are either relatively non-influential or have negligible impact in the KGE performance. $d2$ is the most important soil layer, probably because it is the thickest soil layer where

most of the roots are found, which is expected to exert strong controls on ET. Dsmax, ds and ws are the baseflow-related parameters, interlinked with each other, associated with the third soil moisture layer $d3$, having a higher impact on low flows. We discard a common set of parameters prior to the model calibration based on weighted average of the sensitivity indices of the subbasins. The weights are assigned based on catchment area. Figure 5 shows the influential and non-influential parameters for the entire basin. The total number of model simulations performed is sufficient to achieve the stability of the screening results (see Fig. S3, Supplement). More details on the Morris screening results are given in the Supplement that accompanies this paper (Sect. S2.2).

Figure 6 shows the performance of VIC with respect to KGE in the calibration and validation period for all the subcatchments in the highest order of their catchment size.

**Table 4.** Ranges of percent change, change in flows, and uncertainty (i.e., difference between max and min predicted flow) in extreme and mean annual flows in all the scenarios with respect to the baseline scenario.

| Mean annual extreme | Ba | Ka | Ke | Su | Sa |
|---|---|---|---|---|---|
| **Near Future** | | | | | |
| Change (%) | 2.3 to 5.5 | 1.4 to 4.7 | 1.3 to 2.7 | 4.7 to 10.7 | 2.7 to 4.3 |
| Change ($m^3 s^{-1}$) | 132 to 289 | 62 to 166 | 42 to 77 | 32 to 75 | 27 to 41 |
| Uncertainty ($m^3 s^{-1}$) | 157 | 104 | 36 | 41 | 14 |
| **Far Future** | | | | | |
| Change ( %) | 2.4 to 6.5 | 1.4 to 5.6 | 1.6 to 3.5 | 6 to 15.4 | 3 to 4.7 |
| Change ($m^3 s^{-1}$) | 137 to 347 | 63 to 195 | 51 to 100 | 42 to 109 | 28 to 45 |
| Uncertainty ($m^3 s^{-1}$) | 210 | 132 | 49 | 67 | 17 |
| **All Cropland** | | | | | |
| Change (%) | 2.4 to 1.2 | 1.2 to 8.6 | 2.1 to 5.7 | 6.5 to 20.5 | 5 to 8.5 |
| Change ($m^3 s^{-1}$) | 124 to 496 | 51 to 301 | 67 to 164 | 45 to 147 | 49 to 81 |
| Uncertainty ($m^3 s^{-1}$) | 372 | 250 | 97 | 102 | 32 |
| **All Forest** | | | | | |
| Change (%) | −4 to −14.4 | −2 to −11.4 | −2.6 to −6.6 | −15.8 to −41 | −13.5 to −22 |
| Change ($m^3 s^{-1}$) | −218 to −712 | −85 to −400 | −86 to −190 | −109 to −289 | −131 to −213 |
| Uncertainty ($m^3 s^{-1}$) | 494 | 315 | 104 | 180 | 82 |

| Mean annual flows | Ba | Ka | Ke | Sa | Su |
|---|---|---|---|---|---|
| **Near Future** | | | | | |
| Change (%) | 3.7 to 7.6 | 2.5 to 6.13 | 2.4 to 4.2 | 4.9 to 9.7 | 3.4 to 4.6 |
| Change ($m^3 s^{-1}$) | 21 to 31 | 8.6 to 16 | 5.2 to 7.5 | 34 to 61 | 2.6 to 3.3 |
| Uncertainty ($m^3 s^{-1}$) | 10 | 7.4 | 2.3 | 27 | 0.7 |
| **Far Future** | | | | | |
| Change (%) | 3.4 to 7.9 | 2.12 to 6.5 | 3.4 to 4.6 | 6 to 13.2 | 3.24 to 4.6 |
| Change ($m^3 s^{-1}$) | 19 to 32.6 | 7.3 to 16.8 | 5 to 8.8 | 4 to 8.3 | 2.4 to 3.3 |
| Uncertainty ($m^3 s^{-1}$) | 13.6 | 9.5 | 3.8 | 4.3 | 0.9 |
| **All Cropland** | | | | | |
| Change (%) | 2.8 to 8.5 | 1 to 7.7 | 2.1 to 5.6 | 6 to 16 | 4.1 to 6.6 |
| Change ($m^3 s^{-1}$) | 15.6 to 35 | 3.4 to 20 | 4.7 to 12 | 4.2 to 10 | 3.1 to 5 |
| Uncertainty ($m^3 s^{-1}$) | 19.4 | 16.6 | 7.3 | 5.8 | 2 |
| **All Forest** | | | | | |
| Change (%) | −4.6 to −14.34 | −2.4 to −11.1 | −2.9 to −7.2 | −14.5 to −34.2 | −12 to −18.6 |
| Change ($m^3 s^{-1}$) | −26.2 to −59 | −8.2 to −29 | −6.4 to −15.8 | −10.2 to −21.3 | −9.1 to −13.3 |
| Uncertainty ($m^3 s^{-1}$) | 33 | 20.8 | 9.4 | 11 | 4.2 |

The KGE ranges for the calibration and validation of daily streamflow for all subcatchments are listed in Table S4 in the Supplement. Overall, evaluation result suggests that the model reproduced the observed flows remarkably well with the median KGE values of 0.85, 0.86, 0.82, 0.75, and 0.63 in calibration and 0.77, 0.82, 0.72, 0.60, and 0.59 in validation at Basantpur, Kantamal, Kesinga, Salebhata, and Sundergarh, respectively. However, we observe a relative reduction in the daily KGE values at the smaller subcatchments (Sundergarh and Salebhata) in both calibration and validation periods. The PBIAS values obtained in the calibration period (Fig. S8, Supplement) indicate that the model tends to be more biased (positively) as the catchment size decreases and that the largest catchment, Basantpur, is least biased. The median PBIAS values at Sundergarh and Salebhata are +9 % and +23 %, respectively, in the calibration period and +19 % and +55 % in the validation period. It is to be noted that sub-basins analysed are effected by human intervention, and ob-

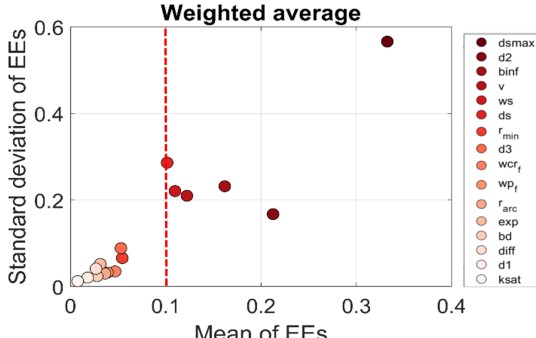

**Figure 5.** Sensitivity indices (mean and standard deviation) of the Morris method for VIC-3L parameters for the Mahanadi river basin (computed based on weighted average of all subcatchments). Parameters, top to bottom, listed on the right side are in ranking order, highest to lowest influential parameters, respectively, based on mean of EEs. The dashed red vertical line is the screening threshold.

served streamflow values are controlled by minor reservoirs and dams which will affect the VIC simulations especially in the smaller subcatchments. Moreover, non-consideration of groundwater recharge and irrigation in VIC can also possibly affect performance at smaller subcatchments. Supplement Figure S7 shows that the models reproduced the daily and monthly flows consistently when compared to the observed flows in both calibration and validation periods.

Figure 6b shows that the distribution of behavioural parameters within their respective variability ranges differs from one parameter to another as well as across subcatchments. The behavioural models at all subcatchments are scattered nearly across the entire range of parameter space for ds and ws, reflecting high effect on modelled streamflow through their interaction with other parameters. Contrarily, behavioural parameter ranges of binf, dsmax, $d2$ and $v$ are relatively constrained across subcatchments, towards either higher, mid or lower values, indicating direct influence of these parameters on the behavioural simulations. For instance, higher values of $d2$ and $v$, lower values of dsmax and mid values of binf resulted in the behavioural model simulations at the smaller subcatchment, Salebhata. Thickness of second soil layer, $d2$, is the most identifiable parameter across all subcatchments.

### 4.2 Baseline scenario performance

We compare the performance of calibrated VIC models in the baseline scenario (using LUH2005) against the validation performance (using the NRSC2005) for the period 2001–2010. The boxplots in Fig. 6a show daily KGE values for the baseline and validation simulations for all subcatchments studied here. The median KGE values for the baseline at Ba, Ka, Ke, Su and Sa are 0.62, 0.64, 0.58, 0.62 and 0.72, respectively. The model performed relatively well at the smaller subcatchments Sa and Su in the baseline, whereas decline in

the performance is observed at subcatchments Ba, Ka and Ke. PBIAS values (Fig. S8, Supplement) indicates that baseline simulations are more biased (negatively) than validation simulations at bigger catchments. The median PBIAS values at Ba, Ka and Ke are −28 %, −29 % and −33 %, respectively. This underestimation can be attributed to the absence of 12 % Barren Ground in the baseline land cover, which is replaced by croplands (4 %), forests (5.02 %), grasslands (4.57 %). The increase in flows due to the increase in cropland is compensated by the decrease in flows due to the increase in forest. Therefore, the underestimation in the simulated flows using LUH2005 may result from the increasing grasslands which increased LAI, thus resulting in an increase in ET and decrease in surface runoff, respectively. Contrarily, a slight positive bias of 3 % is observed at the smallest subcatchment (Sa) in the baseline simulation, compared to +55 % in the validation simulation. KGE values obtained across calibration, validation and baseline periods indicate an overall good performance of the basin as per the existing studies using KGE as a performance metric (Knoben et al., 2019). Overall, baseline land cover map LUH2005 shows comparable model performance against local land cover map NRSC in the historical period with the model being able to capture the seasonality and land use/land cover dynamics while simulating the daily flows.

### 4.3 LULC impacts and uncertainties

Figure 7 shows percent change in annual average of extreme flows (i.e., 95th percentile or higher) for the time 2001–2010 in scenarios NF, FF, All Cropland (CL) and All Forest (F) with respect to baseline scenario for the behavioural models. The range of percent change represents the related uncertainty in model predictions arising from the behavioural model parameters. We observe an insignificant positive change in projected extreme flows in the present (P) scenario despite a major increase, 6 % to 36 %, in croplands replacing forests across four out of five subcatchments (not shown Fig. 7). We observe a prominent increase in the extreme flows at all subcatchments in both future scenarios (NF and FF). The projected change in extreme flows in NF ranges between 1.3 % and 10.7 % across the subcatchments. The median percent change in the NF scenarios at subcatchments Ba, Ka, Ke, Su and Sa are 3.6 %, 2.6 %, 1.8 %, 8.1 % and 3.8 %, respectively. This increase in extreme flows in NF can be attributed to the reduction in forest cover (−20 % to −42 %) at the expense of cropland (+7 % to +48 %) across the subcatchments. Percent increases of slightly higher magnitudes are observed in the FF scenario in response to further increase in croplands. The projected changes in extreme flows in FF ranges between 1.4 % and 15.4 % across the subcatchments. The median percent change in the FF scenario at subcatchments Ba, Ka, Ke, Su and Sa are 4 %, 2.8 %, 2.3 %, 11.3 % and 4.1 %, respectively, in response to reduction in forest cover (−19 to −50 %) at the expense of cropland

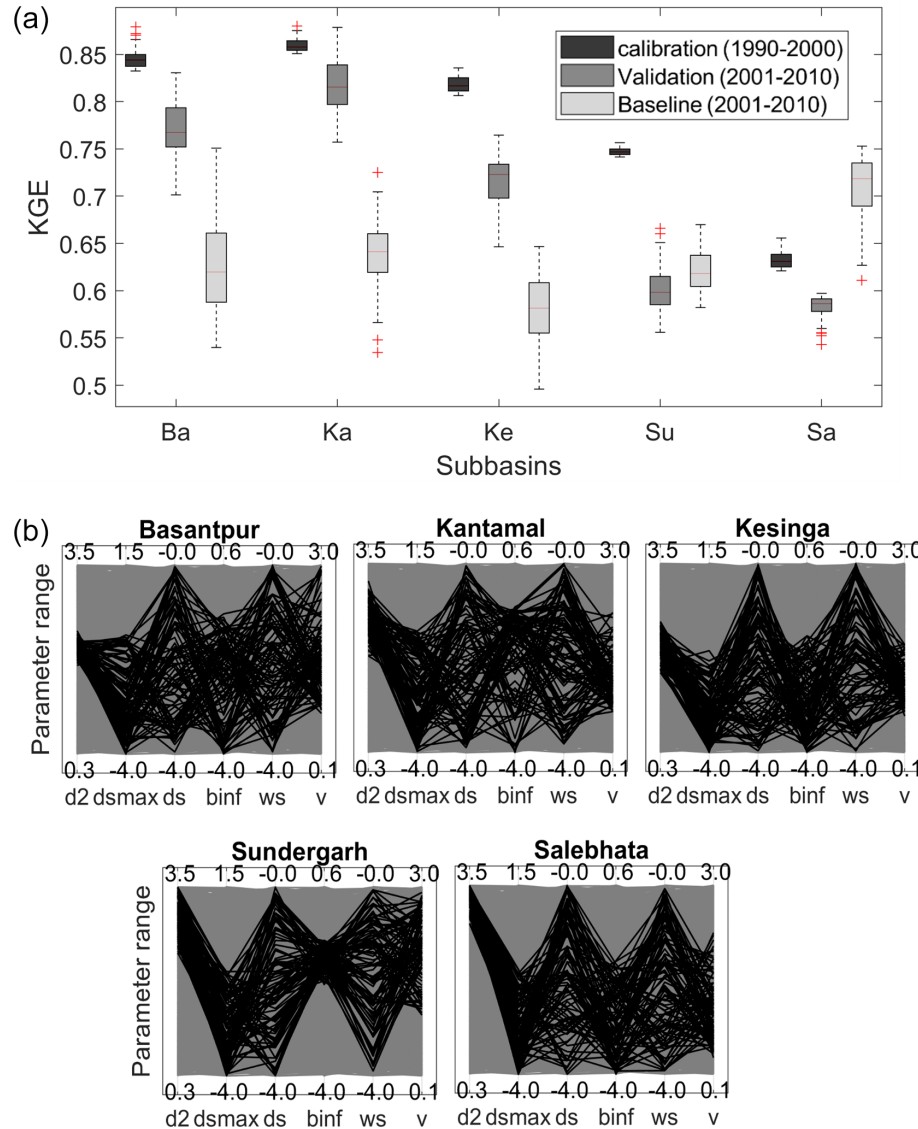

**Figure 6. (a)** Boxplot showing KGE range for calibrated, validated and baseline scenario simulations. **(b)** Parallel coordinate plot representing VIC-3L behavioural parameterization for all subcatchments obtained during model calibration. Lines in black are simulations where KGE lies within top 2 % i.e., behavioural simulations, and lines in grey are non-behavioural simulations. Behavioural KGE values at Ba, Ka, Ke, Su and Sa range from 0.83–0.88, 0.85–0.88, 0.81–0.84, 0.74–0.76 and 0.62–0.66, respectively. Parameters are defined in Table 1.

(+19 % to +54 %) across the subcatchments. As anticipated, maximum percent increases in the extreme flows (1.2 % to 20.5 %) are observed in the hypothetical All Cropland scenario where all forests and grasslands are replaced by cropland and maximum reduction (−2 % to −41 %) observed in the All Forest scenario where all the croplands and grasslands are converted to forests. The projected percent changes in mean annual flows are slightly higher than the extreme flows across all scenarios and subcatchments. The median values in both future (FF and NF) and CL scenarios show slightly higher positive percent change in the range of 3 % to 11 % and higher negative percent change, −5 % to −25 %, in the F scenario.

Maximum increments in extreme flows and annual flows across all scenarios are recorded at the largest subcatchment Basantpur, which are in the range of 194 to 496 and 31 to 35 m$^3$ s$^{-1}$, respectively. The maximum reduction of 712 and 59 m$^3$ s$^{-1}$ is observed in the All Forest scenario at Basantpur. Much less change in terms of magnitudes is observed in the annual flows compared to the extreme flows. This can be explained by the fact that the basin receives approximately 85 % of the total annual rainfall during the monsoon months (June–September). Therefore, with negligible changes occurring during the rest of the year, changes in extreme flows occurring only during the monsoon months are masked out when computed for the entire year. We further computed the

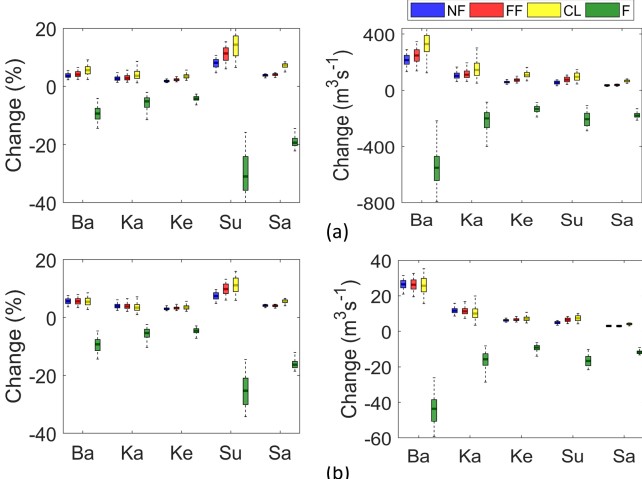

**Figure 7. (a**, left) Percent change in extreme flows (i.e., 95th percentile or higher).(**a**, right) Change in extreme flows (in m$^3$ s$^{-1}$).(**b**, left) Percent change in mean flows. (**b**, right) Change in mean flows (in m$^3$ s$^{-1}$), averaged annually over 2001–2010 in the Near Future (NF), Far Future (FF), All Cropland (CL) and All Forest (F) scenarios with respect to baseline scenario for all the subcatchments. Note that the daily meteorological forcing used in all the model simulations are obtained from the current climatology (i.e., 1990–2010). The results are shown for the behavioural model simulations obtained through calibration.

difference between maximum and minimum values (ranges) of projected extreme flows as a measure of the amount of uncertainty contained in ensemble predictions made using land cover scenarios and multiple (behavioural) parameter sets (Table 4). Uncertainty in Far Future scenario ranges from 17 to 210 m$^3$ s$^{-1}$ across subcatchments. Among all the scenarios, maximum uncertainty is observed in the hypothetical All Forest scenario ($-82$ to $-494$ m$^3$ s$^{-1}$) followed by All Cropland scenario (32 to 372 m$^3$ s$^{-1}$). Overall the uncertainty of hydrological model parameterization is observed at the largest subcatchment Basantpur and decreases with respect to the decrease in the catchment size.

We analysed the water balance components to understand the factors causing changes in the streamflow. Overall, we found that the increase in the mean annual flows is caused by the increment in runoff and reduction in ET across all subcatchments. Positive median changes are observed in runoff (NF, FF and CL), ranging between (2.8 to 14) % and negative changes of ($-4$ to $-37$) % in the F scenario. Negative median changes are observed in ET in scenarios (NF, FF and CL) ranging between ($-1.4$ to $-3.4$) % and positive changes of (1.9 to 7.8) % in the F scenario. Removal of forests decreases the LAI of the natural vegetation and hence decreases ET. Moreover, the removal of forest cover reduces the root water uptake by plants, which increases the water content of the second and third layer of the soil. The top, thin soil layer in the VIC model helps in partitioning the rainfall amount into

direct runoff and the amount entering the soil. Therefore, the increase in the cropland results in more direct runoff, thus reducing the soil moisture content in the first soil layer. The increase in runoff is not significant, despite the occurrence of major deforestation in the future scenarios. This is because the decrease in ET due to forest removal is compensated as increment in croplands also leads to a major increase in ET rates, which is why we do not see a sharp reduction in the ET rates. Negligible changes are observed in baseflow, while slight increase in total soil moisture is noticed across the subcatchments (not shown). The water balance indicates that 15 % to 21 % of precipitation is direct runoff and 64 % to 80 % is ET across all subbasins and all land cover scenarios, whereas negligible baseflow and soil moisture changes are observed. This is probably because the third soil moisture layer in the model does not reach saturation to cause the non-linear baseflow, as precipitation in the basin is highly concentrated in only 3 to 4 months in monsoon, and the rest of the year remains dry.

## 5 Discussions

Performing a comprehensive sensitivity analysis and model calibration enhances the accuracy of hydrological predictions, which subsequently improves the representations of changes in the hydrological regime due to land cover changes. Our SA results are in agreement with existing studies conducted on several basins using VIC, which show binf and $d2$ are the most sensitive parameters (Demaria et al., 2007; Gou et al., 2020; Lilhare et al., 2020; Yeste et al., 2020). Moreover, not all the parameters recommended for calibration by VIC model developers (binf, $d1$, $d2$, $d3$, ds, dsmax and ws) are sensitive to the basin runoff, which is also in line with findings of Bao et al. (2011), Demaria et al. (2013) and Gou et al. (2020) for other basins. For instance, first- and third-layer soil depths ($d1$ and $d3$) are not found sensitive in this study. $d1$ is the thinner topmost soil layer, having not much control on ET and subsurface processes. $d3$ is probably not sensitive as most of the roots are present in the second soil layer, hence not contributing to the soil moisture uptake through the roots. We found that soil properties impose greater control on model performance than the vegetation parameters. However, while varying soil depth influences the ET rates by posing indirect influences on both timing and magnitude of the soil water available for ET, varying root depth and fractions (using our root zone allocation approach) has provided substantial control over the water balance by directly influencing the ET rates, thereby improving KGE (not shown). The weakness in reproducing flows at smaller subcatchments in Mahanadi basin is also reported previously in some studies (Kneis et al., 2014; Mishra et al., 2008).

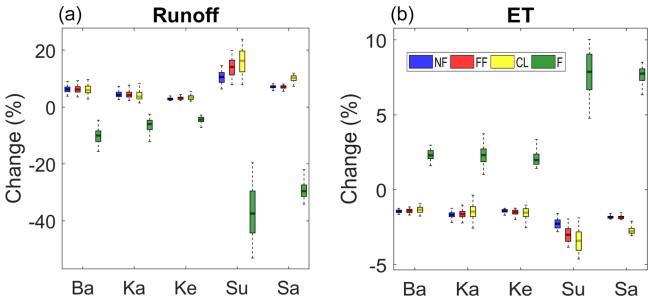

**Figure 8.** Percent change in **(a)** mean runoff **(b)** mean ET averaged annually over the time (2001–2010) in the Near Future (NF), Far Future (FF), Cropland (CL) and Forest (F) scenarios with respect to baseline scenario for all the subcatchments. Note that the daily meteorological forcing used in all the model simulations is obtained from the current climatology (i.e., 1990–2010). The results are shown for the behavioural model simulations obtained through calibration.

LUH2 is a new dataset that is not yet extensively used in basin-scale hydrology. A recent study by Krause et al. (2019) predicted worldwide increment in runoff (67 %) and a variable response of ET across different land use scenarios using LUH2 dataset. The major land cover changes in the future scenarios in Mahanadi basin (as predicted by LUH2) agrees with Behera et al. (2018), wherein they reported a prominent conversion of DBF to croplands in year 2025.

Our findings indicate an increase of 27–496 $m^3 s^{-1}$ in extreme flows and 2.6–35 $m^3 s^{-1}$ in annual mean flows due to deforestation, across the subbasins and scenarios (including the hypothetical cropland scenario). These increasing trends are consistent with other studies in the Mahanadi river basin in India (Dadhwal et al., 2010), neighbouring basins (Das et al., 2018; Kundu et al., 2017) and elsewhere (Abe et al., 2018; Berihun et al., 2019; Cornelissen et al., 2013; Costa et al., 2003). Kundu et al. (2017) found an increase in runoff and decrease in ET due to the expansion in projected agricultural land in Narmada river basin in India. Das et al. (2018) predicted that deforestation, urbanization and cropland expansion in eastern river basins of India in the future would increase runoff and baseflow and decrease ET%. It should be noted that 15 % of the agricultural land in the basin is under irrigation effects; however, this version of VIC (version 4.2.d) does not represent irrigation. Therefore, reduction in ET rates due to conversion of forest to cropland could be compensated by the moisture available due to the irrigation during the non-monsoon season. However, this may not have a significant effect on the assessments of impacts on runoff, especially on extreme flows, because those events are likely to be related to the monsoon season, where the effect of irrigation is minimum.

We found a small change in mean annual discharge as well as in water balance components despite a major change in land cover. Our results correlate well with several research

studies (Ashagrie et al., 2006; Fohrer et al., 2001; Hurkmans et al., 2009; Kumar et al., 2018; Patidar and Behera, 2019; Rogger et al., 2016; Viglione et al., 2016; Wagner et al., 2013; Wilk and Hughes, 2002), wherein they have reported that the impacts of land cover change on water balance components in a large-scale river basin are too small to be detected due to the compensation effects. Wilk and Hughes (2002) showed that removal of large forests led to little or no changes in annual runoff in large heterogeneous catchments in South India. Patidar and Behera (2019) in a recent study in a large river basin in India reported that the conversion of forest to agriculture may not alter the water balance significantly as the impacts on ET and runoff cancel out at the basin scale. The range of these hydrological estimates (Figs. 7, 8 and Table 4) provides more straightforward and explicit quantification of uncertainty than other statistical measures such as variance or interquartile ranges (Her et al., 2019). Our results suggest that even a small set of calibrated models can predict a wide range of flows through different hydrological processes occurring within the basin; therefore, the impacts of uncertainty derived from model parameters on the relative changes cannot be neglected. The uncertainty due to model parameters did not alter the trend of changes in extreme flow, mean annual flow and hydrological components due to land use change in comparison to the baseline simulations. However, a considerable variation is observed especially in the magnitudes of extreme flows simulated for the different land cover scenarios. For instance, the competing interactions among ds and ws led to the varying hydrological processes occurring within the basin, thereby affecting the partition of water in the soil column. Similar conclusions are outlined in Chen et al. (2019b) that the projected monthly and annual flows simulated for different land use scenarios were having significant uncertainty due to model parameterization. In addition, we found that the trends within the scenarios especially in the mean annual flows, runoff and ET are not consistent. For instance, we expect the increase in flows to be more in Far Future scenarios than Near Future, given that the increase in agricultural land in the Far Future is relatively more. However due to different parametrization, some models predicted decrease in Far Future flows relative to Near Future (Fig. 7). This clearly indicates that the impact of land use could be biased when a single model prediction is used, as the impacts could be potentially hidden within simulation uncertainty derived from model (Chen et al., 2019b). Only a small percentage of model simulations (2 %; 100 model simulations) with relatively high daily KGE scores (KGE > 0.8 at 3 out of 5 subcatchments) were used for assessing the impacts, yet significant variations in extreme flow magnitudes and trends (in some cases) are observed. Therefore, selecting models with relatively lower KGE values might have led to larger uncertainty bounds and inconsistent trends in the relative change. Equifinality in hydrological modelling and its influence on hydrological analysis of climate change has been discussed in several studies. However,

its influence on hydrological analysis of land cover change has not been studied enough to provide a clear idea about the contributions of model parameter uncertainty to hydrological projections. Our results thus underline the importance of considering model uncertainty and consequently equifinality while modelling the land cover change impacts.

# 6  Conclusions

In this study an attempt is made to quantify the hydrologic response of the subcatchments of the Mahanadi river basin, owing to different land cover scenarios obtained from the LUH2 dataset, through the implementation of a sensitivity-based calibrated semi-distributed hydrological model. Our findings offer insights into the plausible hydrological scenarios in future at a river basin level, which is a crucial step forward for a developing country in the context of today's increasing focus on integrated water resources management (IWRM) in river basins. Overall, VIC captured the observed daily flows well in calibration, validation and baseline periods across subcatchments. Deforestation at the expense of cropland dominated the land cover change processes across all scenarios and subcatchments, which has led to an increase in the extreme flows and mean annual flows. Analysis of other hydrological components have shown that the increase in flows is caused by the increase in runoff and decrease in ET. The uncertainties due to model parameterization in land use change impacts varies from one subcatchment to another. The uncertainties did not alter the trend of changes when compared to the baseline; however, a considerable variation is observed especially in the magnitudes of extreme flows simulated for the different land cover scenarios. This result suggests a significant constraint on the usage of hydrological models for the variations of extreme flows due to land cover change, even with high KGE values at daily time step as the impacts could be potentially hidden within simulation uncertainty derived from the model parameters. The uncertainties from model parameters thus should be considered in land use change impact assessment for a more robust and reliable analysis, which shall make the land cover change mitigation strategies and water resources management plans more effective.

This study indicates that the recurrent flood events occurring in the Mahanadi river basin might be influenced by the changes in LULC at the catchment scale. However, projected increase in precipitation due to climate change might have more pronounced effect on the streamflow on this basin, especially extreme flows (Asokan and Dutta, 2008; Ghosh et al., 2010; Jin et al., 2018), thereby hiding the hydrological impacts of LULC changes. Future studies shall focus on modelling the combined impacts of climate and land cover changes on hydrology of the Mahanadi river basin, considering the uncertainties from model parameterization, which is currently lacking in many studies.

*Data availability.* The DEM was acquired from Bhuvan, Indian Geo-Platform (https://bhuvan-app3.nrsc.gov.in/data/download/index.php) of Indian Space Research Organisation, last access: 30 November 2021. Values of Unit Hydrograph are obtained from the Variable Infiltration Capacity (VIC) model; Routing: Unit Hydrograph (UH) file (https://vic.readthedocs.io/en/vic.4.2.d/Documentation/Routing/UH/, last access: 30 November 2021). Daily gridded rainfall, maximum and minimum temperature data used in this study can be obtained from the Indian Meteorological Department (IMD) (https://www.imd.gov.in, home page/rainfall and temperature information, last access: 30 November 2021). Wind speed data used in this study can be obtained from NCEP/NCAR reanalysis (Index of/Datasets/ncep (noaa.gov), last access: 30 November 2021) TS1. The source code for VIC-3L version 4.2.d is available from GitHub (https://github.com/UW-Hydro/VIC/releases/tag/VIC.4.2.d). For downloading LUH2 datasets (https://luh.umd.edu/data.shtml), please refer to Hurtt et al. (2020), https://doi.org/10.5194/gmd-13-5425-2020. Observed discharge data are obtained from the Central Water Commission, India (http://www.cwc.gov.in/).

*Supplement.* The supplement related to this article is available online at: https://doi.org/10.5194/hess-25-1-2021-supplement.

*Author contributions.* SN, MARR and RR designed this study. SN performed the model simulations. MARR and RR assisted SN in analysing and discussing the results. SN wrote the article, and MARR and RR commented on the article.

*Competing interests.* The contact author has declared that neither they nor their co-authors have any competing interests.

*Acknowledgements.* This work was funded by the BEMUSED project, which is funded by UK Natural Environment Research Council (NERC; grant number NE/R004897/1) CE1.

*Financial support.* This research has been supported by the Natural Environment Research Council (grant no. NE/R004897/1).

*Review statement.* This paper was edited by Thom Bogaard and reviewed by two anonymous referees.

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

**Remarks from the language copy-editor**

CE1   Please confirm the minor changes.

**Remarks from the typesetter**

TS1   Please provide a link for the last access date and make sure to add reference list entries for each link given in this section.