# Peer review of "Quantifying the impacts of land cover change on hydrological responses in the Mahanadi river basin in India"

_Hydrology and Earth System Sciences, 2021_

## Author Comment (AC1)

**Reviewer 1**

We appreciate the comments and insights provided by Reviewer#1, and below in **bold** include our response to the comments.

**General Overview:**

This is my second review of the study by Naha et al in which the authors attempt to quantify the changes in the hydrological cycle due to changes in the land cover under future climate change scenarios. I am pleased to see that the authors have taken onboard all my suggestions/criticisms of the previous version. Overall, the manuscript has significantly improved, and the results and analysis are appropriate for a regional study of land cover impacts on hydrology. I especially appreciate the inclusion of a comprehensive sensitivity analysis of the model parameters. I only have a few minor comments which are easily addressable.

**We would like to thank the reviewer for this positive feedback. We are happy to know that our proposed changes are being well received by the reviewer. We thank the reviewer for their comments/suggestions on the previous version of our manuscript.**

**Specific comments:**

1. It is interesting to see that soil related parameters are the most important parameters for runoff. A naive question: Are there any links between soil parameters and land cover change? I am not very well-versed in VIC but my question is whether changing landcover also impacts these soil-related parameters?

**In VIC, changing landcover would not impact the soil parameters directly, as all the soil related parameter values are assigned solely based on soil textures.**

2. In Line 110, the authors claim that daily time steps are used in the study. How reliable are daily values derived from climate change scenarios? It would be great if the authors can elaborate on the robustness and relevance of the daily simulations, especially in a climate change study.

**We thank the reviewer for raising this point which we believe may have been caused by lack of clarity in the manuscript. In our study, we only test the impact of land-cover/land-use changes using LUH2 scenarios. In order to isolate these scenarios, we fixed the meteorological forcing using 'current' climate from 1990-2010 available daily from the Indian Meteorological Department. Notice that VIC is commonly employed at daily scales especially when running using the water balance only mode (Gou et al., 2020; Hengade et al., 2018; Hurkmans et al., 2009).**

3. Line 50: Here the authors suggest that the croplands have increased by 82%. I think it would be helpful to have a baseline (82% increase compared to which year?)

**Thanks! We will rephrase this sentence accordingly in our revised manuscript and provide the supporting reference. The proposed revised text is shown below**

**"A recent analysis on global land cover changes using recent satellite remote sensing data (2000–2017) reveals 86% changes in land cover pattern in India, out of which 82% is detected as croplands and a minor 4% as forests (Chen et al., 2019)"**

4. Line 225-230: The authors find that LAI values are in good agreement with a nearby Indian basin. It would be useful to mention the name of the basin here.

**Thanks! We will add the basin name. "Ganga river basin" (Patidar and Behera, 2019)**

5. Line 455: "...the model is able to estimate all the water budget components and maintain proper closure...". This statement is very misleading. Unless, I have missed the validation of the other water balance components, the authors do not yet have evidence to support this statement. Of course, in this study, this is not very important as relative changes are more important, but several other studies have shown how calibrating with only streamflow adversely affects the accuracy of other water balance components. Moreover, hydrological and land surface models close the water and energy balance by construct, so the claim of proper closure is untrue.

**We agree with the reviewer and will remove this statement from our revised manuscript.**

6. There are several grammatical and language-related idiosyncrasies which need to be corrected. I request the authors to rectify them (I only give a few examples here). Evapotranspiration need not be capitalized similar to the other water balance components. Also, Potential Evapotranspiration can be just potential evapotranspiration.

Line 110-115: "...impact studies is limited just with ..." should be "...impact studies are limited just to..."

Line 220: "....are not accounted for SA" should be ...are not accounted for in SA"

Line 480: "...enhances the accuracy for predicting hydrological responses..." could be "...enhances the accuracy of hydrological predictions..."

Line 485: "...are sensitive to this basin..." is not very accurate. Do you mean basin's runoff?

**Thanks, we will correct all grammatical and language-related idiosyncrasies in our revised manuscript.**

---

## Author Comment (AC2)

**Reply to Reviewer 2**

We thank for the comments and feedback provided by Reviewer 2. The responses to all the comments/questions raised by the reviewer are included below in **bold**.

**General Overview:**

The manuscript "Quantifying the Impacts of Land Cover Changes on Hydrological Responses in India" predicts the future impacts of Land Use Land Cover (LULC) change on hydrological regime, with uncertainties, in Mahanadi River basin, India, using Variable Infiltration Capacity (VIC) model. The manuscript is well structured, and the performed modelling experiment is theoretically sound.

The key contribution of the study is in quantifying the parameter uncertainty in predicting the impacts of LULC change on hydrological components. The impact on streamflow is the key focus in the study. My comments and questions on the manuscript are given below.

**We would like to thank the reviewer for this positive feedback**

**Specific comments:**

I feel that inclusion of Basin name in the title would make it more appropriate.

**Thanks! We will include the basin name in the tile: "Quantifying the impacts of Land Cover Change on Hydrological Responses in the Mahanadi River Basin in India"**

Results of the study indicate the importance of parameter uncertainty in assessing the hydrologic impacts of LULC change. In most of the sub-catchments the uncertainty is considerably large, highlighting the importance of considering uncertainty in such assessments. With uncertainty bounds, the quantified impacts are more realistic and useful to decision makers. However, in such studies the input uncertainty, due to uncertain land cover maps, is also important which is not included in the assessment. Categorical and positional uncertainties in LULC maps could increase the total uncertainty, and in some cases, the total uncertainty could be even higher than the net impacts.

**We agree that the uncertain land cover maps could contribute to the total uncertainty. We use future land cover maps from the Land Use Harmonisation 2 (LUH2) database (Hurtt et al., 2011) which consists of six future scenarios (See Table S2). However, we use the information from a single scenario (RCP3. SSP4) because the percentage of land cover change relative to the baseline from other scenarios is either negligible or are comparable to our chosen scenario. Our chosen scenario shows the maximum changes in land cover; hence it was selected as the 'worst case' scenario for our simulations and will likely produce the largest impact.**

It is mentioned that the conversion from forest to agriculture reduces ET (line 26, and 462). However, there may not be very significant net change in ET due to forest-to agriculture conversion. The additional moisture available during non-monsoon seasons due to irrigation could compensate the decreased LAI (due to conversion from forest to agriculture). The impact of such conversion will be only significant during monsoon season, when effect of irrigation is minimum. Since the irrigation was not considered in this study, the impacts on ET due to forest-agriculture conversion may not be very useful. However, this may not affect the assessment of impacts on runoff.

**We thank the reviewer for this comment. Yes, we did not consider irrigation in our study, and we will add a further discussion related to this point in our revised version of the manuscript.**

Authors mentioned that the recurrent flood events in the basin might be due to LULC change (line 31, 579). In my view, the LULC change has a little role in affecting the peak runoffs. With increased precipitation intensity the effect of LULC reduces, therefore during the episode of high precipitation which causes flood may not be affected by small scale LULC changes. Also, the precipitation is much more sensitive to affect runoff as compared to LULC. I believe that if the model is re-run with varying precipitation, instead of keeping it constant (line 337), the impact of LULC change may not be visible.

**We agree with the reviewer, but we believe the addition of experiments with varying precipitation would likely distract readers from the main point of the paper and it is in fact beyond the scope of our study. However, for completeness, we will highlight this point in our revised manuscript.**

The LULC proportions (%) mentioned in Table 3 are not summing to 100% in most of the cases, particularly for 'Salebhata'.

**We thank the reviewer for pointing this out. This is a typo, and we will correct this in our revised version of the manuscript.**

If the uncertainty assessment is included, does the model calibration add any value to the performance? Generally, the model parameters are perturbed between the feasible lower upper bounds by taking some probability distribution. In such case, is there any use of calibrated parameters?

**We believe it does. Model calibration is an important aspect particularly to remove unwanted biases coming from the model structural deficiencies. We believe the fact that we have kept the uncertainty ranges coming from the most influential parameters allows us to evaluate the spread of calibrated model.**

Please expand 'USGC' at line 194.

**Thanks! This will be corrected to "United States Geological Survey (USGS)" in the revised manuscript.**

The line 303 is not clear to me – 'The land cover maps from LUH2 are processed and converted to a LULC map of Mahanadi basin extent showing a single vegetation coverage at each grid cell of 0.25 and further converted to VIC grid size of 0.05 deg'. Does that mean each grid contains only one vegetation?

**Thanks for pointing this out. We believe that the confusion may have been caused by the lack of clarity in the manuscript.**
**No, each grid contains more than one vegetation type, as VIC model maintains heterogeneity within the vegetation types.**
**The LUH2 approach estimates the gridded land use fractions, annually at a resolution of 0.25°. The land use fraction maps are available for each land use type at a resolution of 0.25°. So, we have first obtained LUH fraction maps of different LULC types for Mahanadi basin extent at a resolution of 0.25° and further re-gridded to VIC grid size of 0.05°. Next, to run the VIC model, we have prepared a vegetation parameter file where we included the**

**fractional coverage of all LULC types for each grid cell ensuring that each grid will contain more than one vegetation type.**

**Please note, that there is not a 'single' spatial map available from LUH2 comprising of all LULC classes, which is why we have included figure 3 that shows spatial maps of 'individual' classes.**

What is 'behavioural model'? Please explain.

**There may exist 'equally probable parameter set' or 'behavioural set' that can yield equally good or acceptable model performance known as behavioural models, due to the complex interactions among the model parameters to represent the complex hydrological processes. This is known as equifinality and is considered as one of the main sources of uncertainty in hydrological modelling. We will make it clearer in our revised manuscript.**

Please correct line 462 'Removal of forests at the **expense** of cropland…..'.

**Thanks for the suggestion. We will change this line to "Removal of forests decreases the LAI of the natural vegetation and hence decreases ET".**

---

## Author Response (AR1)

**Response to Reviewers for Manuscript: Quantifying the impacts of land cover change on hydrological responses in the Mahanadi river basin in India**

We appreciate the comments and insights provided by both reviewers. We have gone through the queries, addressed all the reviewer's comments and made all changes as suggested. In addition, we have also made some minor changes. Our detailed responses are presented in here (marked in blue), whilst all changes we have made are highlighted in blue/red in the revised manuscript (track change version).

**Reviewer 1**

**General Overview**

This is my second review of the study by Naha et al in which the authors attempt to quantify the changes in the hydrological cycle due to changes in the land cover under future climate change scenarios. I am pleased to see that the authors have taken onboard all my suggestions/criticisms of the previous version. Overall, the manuscript has significantly improved, and the results and analysis are appropriate for a regional study of land cover impacts on hydrology. I especially appreciate the inclusion of a comprehensive sensitivity analysis of the model parameters. I only have a few minor comments which are easily addressable.

We would like to thank the reviewer for this positive feedback. We are happy to know that our proposed changes are being well received by the reviewer. We thank the reviewer for their comments/suggestions on the previous version of our manuscript.

**Specific comments:**

1. It is interesting to see that soil related parameters are the most important parameters for runoff. A naive question: Are there any links between soil parameters and land cover change? I am not very well-versed in VIC but my question is whether changing landcover also impacts these soil-related parameters?

In VIC, changing land cover would not impact the soil parameters directly, as all the soil related parameter values are assigned solely based on soil textures.

To avoid any further confusion, we have included this information as given below in our methodology section in the revised manuscript with track changes.

**See line 360-364, page 12**

"It is also worth mentioning that running model simulations with different land cover scenarios would not directly impact the soil parameters identified in our chosen behavioural models. That is because all soil related parameter values in VIC are assigned solely based on soil textures"

2. In Line 110, the authors claim that daily time steps are used in the study. How reliable are daily values derived from climate change scenarios? It would be great if the authors can elaborate on the robustness and relevance of the daily simulations, especially in a climate change study.

We thank the reviewer for raising this point which we believe may have been caused by lack of clarity in the manuscript. In our study, we only test the impact of land-cover/landuse changes using LUH2 scenarios. In order to isolate these scenarios, we fixed the meteorological forcing using 'current' climate from 1990-2010 available daily from the Indian Meteorological Department. Notice that VIC is commonly employed at daily scales especially when running using the water balance only mode (Gou et al., 2020; Hengade et al., 2018; Hurkmans et al., 2009).

For clarity in our manuscript regarding not using climate change scenarios, and only applying land cover scenarios, we have included following text in our revised manuscript with track changes.

See Line 352-356, Page 12. Also included in Figure captions for clarity (See Figure 7 and 8).

"Notice that the daily meteorological forcing used in all the model simulations is the same and obtained from the current climatology (i.e., 1990-2010). Here, we focus on identifying the impacts on hydrological responses mainly by applying individual land-cover scenarios. Therefore, any changes observed in the predicted hydrological components will be only attributed to changes in LULC"

We also included texts to further justify employing VIC model on daily scales.

See Line 177-178, Page 6.

"Note that the VIC model is commonly employed at daily scales especially when running with the water balance mode only (Gou et al., 2020a; Hengade et al., 2018; Hurkmans et al., 2009)."

3. Line 50: Here the authors suggest that the croplands have increased by 82%. I think it would be helpful to have a baseline (82% increase compared to which year?)

Thanks! We have updated this text in in our revised manuscript with track changes.

See Line 51-53, Page2.

**"A recent analysis on global land cover changes for the 2000-2017 period (Chen et al., 2019a; IPCC, 2019) revealed 86% changes in land cover pattern in India with 82% detected as croplands and the remaining 4% as forests"**

4. Line 225-230: The authors find that LAI values are in good agreement with a nearby Indian basin. It would be useful to mention the name of the basin here.

Thanks! We have updated this in the revised manuscript with track changes.

See 236-237, Page8.

**"We find the range of MODIS LAI obtained for each LULC type are well in agreement with the LAI values obtained in the nearby Ganga river basin in India (Patidar and Behera, 2019)."**

5. Line 455: "...the model is able to estimate all the water budget components and maintain proper closure...". This statement is very misleading. Unless, I have missed the validation of the other water balance components, the authors do not yet have evidence to support this statement. Of course, in this study, this is not very important as relative changes are more important, but several other studies have shown how calibrating with only streamflow adversely affects the accuracy of other water balance components. Moreover, hydrological and land surface models close the water and energy balance by construct, so the claim of proper closure is untrue.

We agree with the reviewer and have removed this statement from our revised manuscript with track changes.

**Please see Line 480-482, Page 16.**

6. There are several grammatical and language-related idiosyncrasies which need to be corrected. I request the authors to rectify them (I only give a few examples here). Evapotranspiration need not be capitalized similar to the other water balance components. Also, Potential Evapotranspiration can be just potential evapotranspiration.

Line 110-115: "...impact studies is limited just with ..." should be "...impact studies are limited just to..."

Line 220: "....are not accounted for SA" should be ...are not accounted for in SA"

Line 480: "...enhances the accuracy for predicting hydrological responses..." could be "...enhances the accuracy of hydrological predictions..."

Line 485: "...are sensitive to this basin..." is not very accurate. Do you mean basin's runoff?

All the grammatical and language-related idiosyncrasies are corrected (visible with the track changes) in our revised manuscript.

**Reviewer 2**

**General Overview:**

The manuscript "Quantifying the Impacts of Land Cover Changes on Hydrological Responses in India" predicts the future impacts of Land Use Land Cover (LULC) change on hydrological regime, with uncertainties, in Mahanadi River basin, India, using Variable Infiltration Capacity (VIC) model. The manuscript is well structured, and the performed modelling experiment is theoretically sound.

The key contribution of the study is in quantifying the parameter uncertainty in predicting the impacts of LULC change on hydrological components. The impact on streamflow is the key focus in the study. My comments and questions on the manuscript are given below.

We would like to thank the reviewer for this positive feedback.

**Specific comments:**

I feel that inclusion of Basin name in the title would make it more appropriate.

Thanks! We have updated the title in the revised manuscript as follows.

See Line 1-2, Page 1

**"Quantifying the impacts of land cover change on hydrological responses in the Mahanadi river basin in India."**

Results of the study indicate the importance of parameter uncertainty in assessing the hydrologic impacts of LULC change. In most of the sub-catchments the uncertainty is considerably large, highlighting the importance of considering uncertainty in such assessments. With uncertainty bounds, the quantified impacts are more realistic and useful to decision makers. However, in such studies the input uncertainty, due to uncertain land cover maps, is also important which is not included in the assessment. Categorical and

positional uncertainties in LULC maps could increase the total uncertainty, and in some cases, the total uncertainty could be even higher than the net impacts.

We agree that the uncertain land cover maps could contribute to the total uncertainty. We use future land cover maps from the Land Use Harmonisation 2 (LUH2) database (Hurtt et al., 2011) which consists of six future scenarios (See Table S2). However, we use the information from a single scenario (RCP3. SSP4) because the percentage of land cover change relative to the baseline from other scenarios is either negligible or are comparable to our chosen scenario. Our chosen scenario shows the maximum changes in land cover; hence it was selected as the 'worst case' scenario for our simulations and will likely produce the largest impact.

For completeness, we have included this information as below in our methodology section.

See line 334-338, Page 11

"For our study, we have not taken into account the actual uncertainty due to the land cover scenarios. However, the percentage of land cover change relative to the baseline from other LUH2 scenarios is either negligible or are comparable to our chosen scenario. Therefore, our chosen scenario which shows the maximum changes in land cover will likely produce the largest impact."

It is mentioned that the conversion from forest to agriculture reduces ET (line 26, and 462). However, there may not be very significant net change in ET due to forest-to agriculture conversion. The additional moisture available during non-monsoon seasons due to irrigation could compensate the decreased LAI (due to conversion from forest to agriculture). The impact of such conversion will be only significant during monsoon season, when effect of irrigation is minimum. Since the irrigation was not considered in this study, the impacts on ET due to forest-agriculture conversion may not be very useful. However, this may not affect the assessment of impacts on runoff.

We thank the reviewer for this comment. Yes, we did not consider irrigation in our study, and we have added a further discussion related to this point in our revised version of the manuscript.

See line 533-540, Page 18.

"It should be noted that 15% of the agricultural land in the basin is under the irrigation effects; however, this version of VIC (version 4.2.d) does not represent irrigation. Therefore, reduction in ET rates due to conversion of forest to cropland could be compensated by the moisture available due to the irrigation during the non-monsoon season. However, this may not have a significant effect the assessments of impacts on runoff, especially on extreme flows, because those events are likely to be related to the monsoon season, where the effect of irrigation is minimum."

Authors mentioned that the recurrent flood events in the basin might be due to LULC change (line 31, 579). In my view, the LULC change has a little role in affecting the peak runoffs. With increased precipitation intensity the effect of LULC reduces, therefore during the episode of high precipitation which causes flood may not be affected by small scale LULC changes. Also, the precipitation is much more sensitive to affect runoff as compared to LULC. I believe that if the model is re-run with varying precipitation, instead of keeping it constant (line 337), the impact of LULC change may not be visible.

We agree with the reviewer, but we believe the addition of experiments with varying precipitation would likely distract readers from the main point of the paper and it is in fact beyond the scope of our study.

However, for completeness, we have highlighted this point in our revised manuscript.

See line 613-616, Page 20

"However, projected increase in precipitation due to climate change might have more pronounced effect on the streamflow on this basin, especially extreme flows (Asokan and Dutta, 2008; Ghosh et al., 615 2010; Jin et al., 2018), thereby hiding the hydrological impacts of LULC changes."

The LULC proportions (%) mentioned in Table 3 are not summing to 100% in most of the cases, particularly for 'Salebhata'.

We thank the reviewer for pointing this out. This is a typo, and we have corrected this in our revised version of the manuscript.

Please note that we have "accepted the changes" in this case.

See Table 3, Page 31.

If the uncertainty assessment is included, does the model calibration add any value to the performance? Generally, the model parameters are perturbed between the feasible lower upper bounds by taking some probability distribution. In such case, is there any use of calibrated parameters?

We believe it does. Model calibration is an important aspect particularly to remove unwanted biases coming from the model structural deficiencies. We believe the fact that we have kept the uncertainty ranges coming from the most influential parameters allows us to evaluate the spread of calibrated model.

Please expand 'USGC' at line 194.

Thanks! We have corrected this to "United States Geological Survey (USGS)" in the revised manuscript.

See Line 201, Page 7.

"United States Geological Survey (USGS)."

The line 303 is not clear to me – 'The land cover maps from LUH2 are processed and converted to a LULC map of Mahanadi basin extent showing a single vegetation coverage at each grid cell of 0.25 and further converted to VIC grid size of 0.05 deg'. Does that mean each grid contains only one vegetation?

Thanks for pointing this out. We believe that the confusion may have been caused by the lack of clarity in the manuscript.

We have now mentioned this in a clearer way in our revised version.

See line 310-315, Page 10.

"The land use fraction maps are available for each land use type at a resolution of 0.25°. So, we have first obtained LUH fraction maps of different LULC types for Mahanadi basin extent at a resolution of 0.25° and further re-gridded to VIC grid size of 0.05°. Next, to run the VIC model, we have prepared a vegetation parameter file where we included the fractional coverage of all LULC types for each grid cell ensuring that each grid will contain more than one vegetation type."

What is 'behavioural model'? Please explain.

See line 72-74, Page 3.

"There may exist 'equally probable parameter set' that can yield equally good or acceptable model predictions (also known as behavioural models) which are identified due to the complex interactions among the model parameters to represent the complex hydrological processes".

Please correct line 462 'Removal of forests at the expense of cropland.....'.

Thanks for the suggestion. We have changed this line to "Removal of forests decreases the LAI of the natural vegetation and hence decreases ET".

See Line 487, Page16.

"Removal of forests decreases the LAI of the natural vegetation and hence decreases ET."

**Some other changes that we have made in the revised version includes:**

- 1. Color coding in *Figure 5* in revised manuscript and *Figure S2, S3* in supplementary file has been changed for better interpretation of the result. Also, the captions of these figures are re-written in a clearer way (visible with track changes).
- 2. We have added one extra sentence in the Results section as given below. Although these uncertainty values can be found in Table 4. We believe it will be easier for the readers to follow, if mentioned in text.

"Uncertainty in far future scenario ranges from 17 to 210 cumecs across subcatchments. Among all the scenarios, maximum uncertainty is observed in the hypothetical 'All Forest' scenario (-82 to -494 cumecs) followed by 'All Cropland' scenario (32 to 372 cumecs)"

---

## Author Response (AR2)

**Response to the editor for Manuscript: Quantifying the impacts of land cover change on hydrological responses in the Mahanadi river basin in India**

We appreciate the minor technical corrections provided by the editor. We made all changes as suggested and marked in bold are our responses.

replace cumecs with m3 s-1
- check also all tables and figures on correct unit notations (m3/s should be m3 s-1, etc)

- I noted LUH2 is 3-4 times written as LUH, check if that is correct or that is always should be LUH

**All LUH in text and figure captions are changed to LUH2.**

- Line 513: "agrees" > "agree"
- Line 512 - 515: sentence somewhat hard to follow, consider simplifying

"The major changes occurring in Mahanadi in the future scenarios as predicted by LUH2 agree with a recent study by Behera et al., (2018) in the same basin, wherein they found a prominent conversion of DBF to croplands in year 2025 relative to year 2005."

**This sentence has been changed as follows.**

**"The major land cover changes in the future scenarios in Mahanadi basin (as predicted by LUH2), agrees with Behera et al., (2018), wherein they reported a prominent conversion of DBF to croplands in year 2025. "**

- Line 529: "this may not have a significant effect the assessments of impacts on runoff" here is a word missing after "effect"
- Line 533: "Our results correlates" > "Our results correlate"
- Line 542: "the impacts on ET and runoff cancels out" > "the impacts on ET and runoff cancel out"
- Line 576: LUH2 instead of LUH

**Thanks. All these changes have been made in the revised version of the manuscript.**